# The Supporting Role of Combined and Sequential Extracorporeal Blood Purification Therapies in COVID-19 Patients in Intensive Care Unit

**DOI:** 10.3390/biomedicines10082017

**Published:** 2022-08-19

**Authors:** Federico Nalesso, Federica L. Stefanelli, Valentina Di Vico, Leda Cattarin, Irene Cirella, Giuseppe Scaparrotta, Francesco Garzotto, Lorenzo A. Calò

**Affiliations:** 1Nephrology, Dialysis and Transplantation Unit, Department of Medicine, University of Padua, 35128 Padua, Italy; 2Unit of Biostatistics, Epidemiology and Public Health, Department of Cardiac Thoracic Vascular Sciences and Public Health, University of Padova, 35128 Padova, Italy

**Keywords:** acute kidney injury (AKI), kidney replacement therapy (KRT), extra-corporeal organ support (ECOS), continuous kidney replacement therapy (CRRT), COVID-19, SARS-CoV-2, high cut-off (HCO) membranes, hemoperfusion (HP), cytokines release syndrome (CRS), extracorporeal membrane oxygenation (ECMO)

## Abstract

Critical clinical forms of COVID-19 infection often include Acute Kidney Injury (AKI), requiring kidney replacement therapy (KRT) in up to 20% of patients, further worsening the outcome of the disease. No specific medical therapies are available for the treatment of COVID-19, while supportive care remains the standard treatment with the control of systemic inflammation playing a pivotal role, avoiding the disease progression and improving organ function. Extracorporeal blood purification (EBP) has been proposed for cytokines removal in sepsis and could be beneficial in COVID-19, preventing the cytokines release syndrome (CRS) and providing Extra-corporeal organ support (ECOS) in critical patients. Different EBP procedures for COVID-19 patients have been proposed including hemoperfusion (HP) on sorbent, continuous kidney replacement therapy (CRRT) with adsorbing capacity, or the use of high cut-off (HCO) membranes. Depending on the local experience, the multidisciplinary capabilities, the hardware, and the available devices, EBP can be combined sequentially or in parallel. The purpose of this paper is to illustrate how to perform EBPs, providing practical support to extracorporeal therapies in COVID-19 patients with AKI.

## 1. Introduction

On 11 February 2020, the WHO announced a new and complex disease caused by a new Coronavirus (CoV), the SARS-CoV-2 [1], and its infection named “COVID-19”, which soon resulted in a pandemic for its very high diffusion capacity [2]. Currently, the emergence of variants of the virus continues to fuel the infection in the world population, highlighting the problem of population immunization and reinfection. Symptomatic and asymptomatic individuals may contribute to up 80% of COVID-19 transmission in the general population, challenging the health systems of all countries to an unsustainable long-term use of human, logistic, and economic resources. The need to rationalize resources is vital to ensure the best treatment in COVID-19. The pathogenesis of SARS-CoV-2 pneumonia produces an excessive immune-reaction depending on the host, determining in some cases a cytokines release syndrome (CRS) followed by extensive tissue damage and dysfunctional coagulation, as documented since the first disease reports [3,4]. The term “MicroCLOTS” (microvascular COVID-19 lung vessels obstructive thrombo-inflammatory syndrome) also well describes the lung viral injury [5].

In details, activated leukocytes produce IL-6 that acts on a large number of cells and tissues, stimulating the production of acute phase proteins with a proinflammatory effect. Recently, the role of endotoxin, both as a component of superinfections and as a result of translocation from the gastrointestinal tract injured by the virus, has been postulated. From these it is clear how the COVID-19 inflammation is complex [6,7], involving all organs. The COVID-19 clinical spectrum varies from asymptomatic forms to respiratory failure needing mechanical ventilation in intensive care units (ICU), to multiorgan and systemic manifestations such as sepsis and septic shock for concomitant superinfections during hospitalization [8,9]. During multiple organ dysfunction syndromes (MODS), acute kidney injury (AKI) is emerging, worsening patients’ outcomes. AKI ranges from mild proteinuria [10] to acute progressive forms and represents a marker of MODS and disease severity, requiring kidney replacement therapy (KRT) in up to 20% of patients, confirming the relationship between alveolar and renal tubular damage in acute respiratory distress syndrome (ARDS) [11] due to organ cross-talk. In COVID-19, the possible use of cytokine removal techniques [12] appears to be useful in selected cases. Therefore, in AKI, the extracorporeal purification therapy, for both renal purification and cytokine removal, results in being highly effective [13], while the use of extra-corporeal membrane oxygenation (ECMO), invasive mechanical ventilation and continuous kidney replacement therapy (CRRT) could also contribute to cytokine generation, causing further inflammation that is deleterious to these patients. In this setting, it is very important to investigate the early AKI detection via specific markers of kidney injury [14,15,16,17] to treat patients early; prevent further AKI; and control the volume overload that may deteriorate pulmonary gas exchange, requiring mechanical ventilation. To date, there is no specific antiviral treatment and monoclonal antibodies recommended for COVID-19, consequently the medical treatment is symptomatic, and oxygen therapy represents the first step for addressing respiratory impairment. Of particular note for the antiviral use is that remdesivir use in COVID-19 excluded patients with stage 4 CKD or those requiring dialysis (i.e., eGFR < 30 mL/min/1.73 m^2^). Non-invasive (NIV) and invasive mechanical ventilation (IMV) may be necessary in cases of respiratory failure refractory to oxygen therapy [18]. At the moment, the therapeutic strategies are only supportive and directed to systemic inflammation reduction to stabilize organ function and reduce the requests for intensive cares to save ICU resources [19]. Extracorporeal blood purification therapies (EBP) have also been proposed as useful approaches to remove cytokines in patients with sepsis [20] and could potentially be beneficial in COVID-19 [21], preventing CRS-induced organ damage [22]. COVID-19 patients, in fact, develop severe forms of MODS that may not be adequately supported only by pharmacologic therapies, and EBP can have a potential role in the so called extra-corporeal organ support (ECOS) [23]. In case of single organ failure, heart, lungs, kidneys, and liver can be partially replaced or at least sustained using specific extracorporeal devices. This support can be also provided in a more complex setting of multiple organs failure using combined or sequential EBP. A single form of ECOS may be required, but multiple organ support therapy (MOST) [24] is currently a feasible approach with the combination of EBPs to provide a personalized ECOS. In critical patients, it is recognized that severe renal dysfunction is a typical syndrome requiring kidney replacement therapy (KRT) and during MODS, ECOS is not seen as an extraordinary or particularly aggressive technique, although the perceived complexity depends on local resources, experience, and the possibility to provide a multidisciplinary approach. Much of the current experience on MOST has been gained upon the KRT connected to ECOS. The evolution of KRT in CRRT combined or in sequential modality with special devices can better support patients under ECMO, extracorporeal carbon dioxide removal, or hemoperfusion (HP). In this view, physicians should be familiar with the concept that EBP represents today an important strategy in critically ill patients, playing an important role in the COVID-19 management. In EBP, HP, CRRT with adsorbing capacity, or the use of high cut-off (HCO) membranes are effective therapies. Depending on the local experience, it is possible to combine EBP sequentially or in parallel to provide ECOS according to clinical needs. This paper describes the most common EBPs in order to provide effective support in their management.

## 2. The Extracorporeal Blood Purification (EBP)

### 2.1. Hemoperfusion (HP)

Hemoperfusion refers to the circulation of anticoagulated blood through an extracorporeal circuit with a disposable cartridge adsorbing specific molecules. The HP circuit is simpler (Figure 1) than that used for hemodialysis but, due to a direct contact between blood and sorbent, requires adequate anticoagulation and a biocompatible sorbent. In the past, charcoal has been extensively used for its high adsorbing capacity, especially for relatively hydrophobic and low molecular weight solutes retained in the case of kidney or liver failure. Synthetic polymers have been made available with remarkable adsorption capacity due to the widened pores on the granules surface with minor size exclusion impact [25]. These devices have demonstrated efficiency in removing poisons, bilirubin, cytokines, and endotoxin. They are divided into selective (e.g., polymyxin B in hemoperfusion) and non-selective types (e.g., CytoSorb) [26]. The selectivity may be important for treatment, as non-selective cartridges cannot adsorb endotoxins for their cutoff point (~60 kDa) below the endotoxin molecular weight (~100 kDa) [27]. In clinical practice, the HP can play an important role in the treatment of patients suffering from initial CRS with no severe organ damage. This approach could therefore limit the severity of organ damage, preventing the progression from dysfunction to organ failure with the need for artificial organs use such as mechanical ventilation, KRT, cardiac support devices, and ECMO. In this simple configuration, it is possible to perform a single treatment to remove cytokines by HP using devices such as CytoSorb, HA-330 cartridge, and Toraymyxin (Figure 1).

CytoSorb (CytoSorbents Corporation, Princenton, NJ, USA) is a developed polystyrene-based hemoadsorber with non-selective capacity. It is composed of biocompatible porous polymer beads in polystyrene divinylbenzene copolymer with a biocompatible polyvinylpyrrolidone coating, which acts like tiny sponges removing substances from whole blood based on pore capture and surface adsorption. Molecules ranging from 5 to 60 KDa can be trapped in the beads channels and pores and are permanently removed. CytoSorb can be a solution to treat a cytokine storm and inflammation as cytokines and inflammatory mediators fall within this size spectrum. Cytosorb is an option to support and treat COVID-19 [23,28] in ICU with the objective of reducing the “cytokine storm” [29], receiving the FDA Emergency Use Authorization on 10 April 2020. This device has been routinely used in clinical practice in the EU for other conditions where an excess of cytokine occurs. According to its absorptive characteristic, this device can also be integrated into a bypass circuit in the ECMO or CRRT (combined therapy). The device duration is up to 24 h and its efficacy is concentration dependent. The standard treatment protocol in COVID-19 involves performing HP using one device every 12 h on day 1, then one device for 24 h on day 2, and eventually an additional device for 24 h on day 3. The recommended blood flow rate is 150–500 mL/min (maximum flow 700 mL/min) with a minimum of 100 mL/min. It has to be highlighted that flow rates below 150 mL/min may be required due to the catheter access limitations, but higher caution should be used with low flow rates due to the increased potential for device clotting. Focal points of this treatment are the need for anticoagulation with a recommended aPTT between 60 and 80 s (or ACT of 160–210 s) and the treatment duration that requires the device to operate for at least 12 h with the goal of 24 h. In COVID-19 patients, a loading dose of heparin of 50–70 IU/kg and a maintenance dose of 15–20 IU/kg/h should be evaluated according to the patient’s clinical condition and bleeding risk. Patients undergoing ECMO should be anticoagulated according to standard clinical practice for those procedures. Definitely, the blood flow can be set in wide range (100–700 mL/min) according to the central venous catheter (CVC) performance and the device use in HP alone or combined with other treatments (CRRT or ECMO). In order to achieve an effective purification with all these modalities of blood purification, at least a standard temporary 12 F double lumens CVC catheter for hemodialysis that can guarantee a blood flow up to 300 mL/min is required.

Summarizing, critical points of this procedure are the patient’s blood coagulation state and the maintenance of the patency of the extracorporeal circulation over time, despite the problems linked with the vascular access and the need for pronation and supination of ICU patients.
-The Jafron HA cartridge (Jafron Biomedical Company, Zhuhai, China) belongs to the non-selective group of non-ionic styrene divinylbenzene copolymers and consists of several types of cartridges (HA-130, HA-230, and HA-330). The cartridges contain neutro-macroporous resin-adsorbing beads. The average diameter of the resin beads is 0.8 mm, ranging from 0.6 to 1.18 mm. The pore size resin distribution determines the range of molecules removal: 500 D–40 kD in HA130, 200–10 kD in HA230, and 500–60 kD in HA330, which has the possibility to remove endogenous and exogenous materials: hydrophobic or protein-bound exogenous substances, cytokines, protein bound uremic toxins, middle uremic toxins, free hemoglobin, and myoglobin [30]. The removal of a wide spectrum of molecular weights is due to differences in resin pore size that make them applicable in settings varying from reduction of uremic symptoms in chronic hemodialysis (HA-130) and the treatment of paraquat and organophosphorus poisoning (HA-230), to modulation of severe inflammatory syndrome (HA-330 and its evolution HA-380) [31,32,33,34], allowing its use in COVID-19. The HP requires a blood flow rate of 100–250 mL/min; the anticoagulation and the therapy duration is similar to that required by CytoSorb (one device every 12 h in the first day, then 24 h in the second and third day, even if other protocols of use are described) [21,31,35]. The clinical experience with the Jafron HA cartridges is mainly limited to China and the counterpart of HA-330/HA-380 is the equally non-selective extracorporeal cytokine adsorber CytoSorb that is clinically available from 2011 and is currently approved for the extracorporeal adsorption technique in Europe. The advantages and disadvantages appear to be similar to those of other styrene divinylbenzene copolymers HP devices.-Toraymyxin (Toray Industries Ltd., Tokyo, Japan) is an extracorporeal hemoperfusion cartridge designed to remove blood endotoxin and is indicated for patients with sepsis or septic shock caused by suspected Gram-negative bacterial infection or endotoxemia. This device is composed of polymyxin B-immobilized on polystyrene derivative fibers [36]. Polymyxin B is able to bind endotoxins selectively. In the device, Polymyxin B is covalently bound to the fibers, preserving them from systemic toxic effects. More than 150,000 patients treated with this innovative therapy demonstrate the safety and effectiveness of Polymyxin B hemoperfusion therapy. With an endotoxin removal capacity of 640,000 endotoxin units, the high affinity binding may remove up to 90% of circulating endotoxin after two hemoperfusion treatments (2 h of HP treatment repeated after 24 h) [37]. Although Polymyxyn-B Hemoperfusion (PMX-HP) therapy was designed to adsorb endotoxins [38], other mechanisms of immunomodulation have been demonstrated resulting from the direct adsorption of inflammatory mediators, cytokines, and the activated monocytes and neutrophils apheresis [39] with positive impact on sepsis [35,40]. On 14 April 2020, the FDA (US Food and Drug Administration) and Health Canada approved Toraymyxin to treat COVID-19 patients suffering from septic shock. Toraymyxin has successfully been used for treatment in COVID-19 patients in the USA, Japan, and Italy. Toraymyxin can neutralize an endotoxin and remove activated immune cell by apheresis decreasing the CRS, improving lung function, and facilitating the weaning from ventilator in patients who developed severe ARDS. The recommended protocol for HP is one session of HP of 2 h (Blood flow of 100 mL/min, maximum blood flow 120 mL/min) repeated after 24 h. Focal points of this treatment are the need for anticoagulation with a desired aPTT between 60 and 80 s and the duration of the treatment that requires the device to operate for 2 h with a reduced risk of bleeding compared to a treatment that lasted for 24 h. Moreover, a 2 h HP treatment per day allows to schedule the therapy and avoid dysfunctions of the extracorporeal circulation due to patients’ position (pronation and supination) or the CVC dysfunctions by the patient’s displacement. Critical points are patient’s blood coagulation and the maintenance of the extracorporeal circulation patency over time despite the vascular access problems and the need for patients’ pronation and supination. Compared to more prolonged treatments requiring anticoagulation, the anticoagulation window is only 2 h for 2 consecutive days, allowing for Toraymyxin use even in patients with a relative bleeding risk.

### 2.2. The High and Middle Cut-Off Filter for Hemodialysis

When the patient’s inflammatory state determines a worsening in the clinical conditions with MOFS, EBP can be addressed in a combined way to remove cytokines and provide KRT reaching both cytokine removal and renal replacement. The use of medium cut-off (MCO) or high cut-off membranes (HCO) in diffusion is simple and easy to perform. HCO filters allow cytokines and other inflammatory factors to be removed [41], ensuring at the same time the renal purification [42]. MCO filters provide a lower removal spectrum of higher molecular weight molecules so that during CRS it is preferable using HCO devices only that appear to have relevance for the removal of cytokines and inflammatory mediators also during AKI requiring KRT. Due to their high cut-off, these filters should be used in diffusive modality only, while convection (hemofiltration or hemodiafiltration) could cause a clinically significant albumin loss [43]. These filters can be used in continuous diffusive modality with a regional citrate anticoagulation (RCA) performing the so called high cut-off-continuous veno-venous hemodialysis in regional citrate anticoagulation (HCO-CVVHD-RCA, Figure 2). A protocol for the Septex set use in HCO-CVVHD-RCA was reported [43] with a positive impact both on the clinical conditions of COVID-19 patients and circuit patency for the entire life of the device (72 h). The possibility of using concentrate citrate solution (4% trisodium citrate) as anticoagulation makes HCO-CVVHD-RCA easy to perform, allowing lower blood flows (90–100 mL/min) and decreasing the number of alarms and circuit clotting due to CVC dysfunctions or patient’s mobilization for the prono-supination.
-Septex (Baxter, Round Lake, IL, USA) is a high cut-off filter with a polyarylethersulfone membrane of 1.1 m^2^ for the use in CVVD (continuous veno-venous hemodialysis). According to its characteristics, Septex can be used in RCA with 4% citrate solution to decrease the pre-blood pump infusion, limiting convection to less than 200 mL/h to avoid significant albumin losses as the device allows a maximal convection of 500 mL/h. The resulting HC-CVVHD-RCA treatment is able to remove cytokines and inflammatory mediators and provide a renal replacement therapy by diffusion the ensuring circuit and filter patency even at low blood flows (80–120 mL/min) until 72 h, which is the device expiration, as per the manufacturer’s instructions [44]. The use of sodium citrate does not expose the patient to any risk of bleeding as anticoagulation is limited at the extracorporeal circuit where it determines an antinflammatory effect on the circuit and membrane surface [45].-Emic-2 (Fresenius Medical Care, Bad Homburg, Germany), is a high cut-off filter with Fresenius polysulfone membrane of 1.8 m^2^. This device is used in CVVHD. Anticoagulation can be performed in RCA with high citrate concentration solution. In this technique, the required blood flow for purification is lower, thus it may be set low enough (100–200 mL/min) to prevent extracorporeal circulation malfunction in the case of CVC dysfunction, not allowing higher blood flow or sometimes inadequate blood flow due to patient mobilization.

With HCO-CVVHD-RCA, both Septex and EMIC-2 are used in CVVHD, while they are excluded from the use in continuous veno-venous hemofiltration (CVVH) due to possible albumin losses. In fact, with a filter with MCO or HCO, in the presence of convective therapies, such as CVVH and CVVHDF, the convective flow can cause a significant loss of albumin for its entrainment through the pores of these membranes, while this fact is not present in diffusive treatments such as CVVHD also using MCO or HCO. Furthermore, the CVVHD treatments has low filtration fractions (FF), as it is based on diffusion, only allowing the use of relatively low blood flows while ensuring the septic depurative dose [46]. Low FF together with sodium citrate anticoagulation results in the lowest risk of clotting. In case of unavailability of these devices to treat AKI and CRS simultaneously, it is possible to provide CRRT with standard membranes to which, according to the technical specifications of the CRRT monitor, the HP can be combined in series (HP + CRRT), or sequentially (HP → CRRT), (Figure 3); if strictly necessary, it is possible to perform the HP and CRRT also in parallel using two distinct vascular accesses (Figure 4).

### 2.3. The Membrane Adsorption

Further evolution of CRRT treatment involves the use of membranes able to remove cytokines and endotoxins. These treatments, operated by devices with a membrane characterized by surface adsorbing capacity, can be effective in the treatment of COVID-19 patients with a high blood level of endotoxin for superinfections [7] or for its translocation from the gastrointestinal tract [6,47], which is not otherwise removed with other standard devices for CRRT. The advantage of this technology concerns the possibility of providing a KRT removing at the same time cytokines and endotoxin; therefore, these types of devices (CRRT filter with a specific membrane) are addressed to AKI patients in which it is necessary to also treat CRS and sepsis. In fact, the possibility of combining CRRT with the removal of endotoxin plays a crucial role in the treatment of AKI COVID-19 patients, in particular when it is not possible to provide HP treatments for endotoxin.
-oXiris (Baxter, Round Lake, IL, USA) is a filter in an AN69-based membrane with a treated surface. In detail, the membrane is composed of acrylonitrile and sodium methallyl-sulfonate-copolymer and as a surface treatment agent polyethyleneImine (PEI) and heparin. This device is intended for use in CVVH, CVVHD, and CVVHDF (continuous veno-venous hemodiafiltration), with or without heparin use, but it can be also used in RCA. This latest Acrylonitrile-69 (AN69-based generation) filter presents unique features as, due to the high concentration of PEI in the inner membrane surface, it presents an increased potential to adsorb endotoxins from the blood and, in addition, the immobilized heparin on the membrane surface gives an anti-thrombogenic property [48]. Due to these characteristics, this filter has been shown to adsorb endotoxin and cytokines [49,50]. EBP with this device showed to be effective with no adverse events on serum IL-6 level reduction, attenuation of systemic inflammation, multiorgan dysfunction improvement, and reduction in ICU mortality rate [48]. In vitro, oXiris displayed a similar adsorption effect for endotoxins of those by Toraymyxin and CytoSorb for the removal of most cytokines and other inflammatory mediators [51]. Early CRRT initiation with this filter in critically ill COVID-19 [48] patients provided a decline in inflammatory markers and prevented multi-organ dysfunction from “cytokine storm” [52]. The versatility of this device allows to prescribe the most indicated CRRT treatment (CVVH, CVVHD, and CVVHDF) to choose the anticoagulation modality according to patient’s bleeding risk and to reach the depurative dose based on the patient’s body weight and clinical need. In order to keep the adsorbing capacity higher, it is recommended to replace the device every 24 h to overcome the limitation of membrane surface saturation over time. Advantages of this device are the possibility of having KRT and the endotoxin and cytokines removal simultaneously. The chance of using sodium citrate as an anticoagulant allows to minimize the device clotting. Disadvantages are the presence of heparin on the device membrane, which contraindicates its use in patients with heparin-induced thrombocytopenia (HIT) type II [53]. Moreover, despite the RCA, the high FF required by some CVVH treatments such as in heavy weight patients and CVC dysfunction, could lead to coagulation of the device, as in the case of reduced blood flows due to CVC dysfunctions or patient mobilization.

Regarding the role of CRRT with standard filters, they provide KRT, and consequently an indirect organ support by treating the fluid overload and compensating the electrolyte and acid-base balance. As described for oXiris, special filters with high absorptive capacity allow to treat the patient’s inflammatory state, also reducing endotoxin levels, but their use is not always available in all ICUs. Valid alternatives are filters which, although classically addressed to renal purification, can play a minor role in purifying from cytokines and other inflammation factors. The final efficacy of these devices is certainly lower than those primarily addressed to cytokines removal, but they can be a valid support in those care settings where it is not possible to provide combined, sequential, or single treatments with specific devices for economic and human resources reasons.

### 2.4. The Polyacrylonitrile and Polymethyl-Methacrylate Membranes

The use of filters in polyacrylonitrile (PAN) or polymethyl-methacrylate (PMMA) can enhance the cytokines removal by adsorption [54,55]. In particular, AN69 surface-treated membrane (AN69ST) is a derivative of AN69 (native AN69) [56] prepared by the surface treatment with polyethyleneimine (PEI). AN69 membrane adsorbs cytokines via ionic binding between its sulfonate group and the amino group on the cytokine molecules surface; in addition, its hydrogel structure allows the adsorption to this membrane within the bulk layer, thereby exhibiting a high adsorption capacity more than polysulfone membranes [57]. The CRRT use, only apparently and reductively aimed to the renal replacement therapy, if administered with these membranes [26], can be considered as a broader treatment to treat COVID-19 patients with AKI and CRS, although with more limited adsorption. In general, for all devices, the timing and the renal dose for CRRT to treat AKI should be according to KDIGO guidelines [58]. Furthermore, given the peculiarity of COVID-19 patients, the depurative dose administered should be guaranteed at least at 35 mL/kg/h as there is no evidence that higher doses may have an impact on survival [46]. In Table 1, all devices for EBP are shown for comparison.

## 3. Discussion

There is currently no specific therapy for SARS-CoV-2 infection universally recognized and standardized. The adopted treatments and the results on the efficacy of their protocols are not yet fully available. The variability of the inflammatory state severity induced by SARS-CoV-2 determines a broad spectrum of disease ranging from asymptomatic to severe forms of MOFS requiring EBP. Nowadays, ECOS is not seen as an extraordinary or particularly aggressive technique in MODS patients in ICU, although the perceived complexity depends on local resources, experience, and the possibility to provide a multidisciplinary approach to the patients. The lack of knowledge in the field of blood purification and the fear in the management of extracorporeal circulation could lead to a lack of use of EBP, especially in those clinical settings with less experience in the extracorporeal blood circulation and circuit anticoagulation. In our experience, the more the multidisciplinary team is prone to perform EBPs, the more treatments are performed safely thanks to the experience gained and the introduction of management protocols. The possibility to use specific extracorporeal purification devices for the removal of cytokines, inflammatory mediators, endotoxins, and for the renal replacement therapy allows to support COVID-19 patients with one or multiple EBP [59] to reach ECOS [20] to integrate with the medical therapy and the ICU (Figure 5). According to the patient’s clinical condition, treating physicians can apply a tailored EBP configuration, regularly assessing the patient for CRS, AKI, and MOFS that can require devices such as ECMO or ECCO_2_-R.

The SARS-CoV-2 pathogenesis is based on systemic inflammation with organs damage, thus it seems useful to monitor patients for organ dysfunction to treat with EBP to reduce systemic inflammation [36] before the organ failure [23,48]. In MODS, the application of EBP may lead to clinical stabilization, avoiding progression towards MOFS. The treatment type is based on the presence or absence of MODS and the local experience in providing these treatments. In the case of the presence of inflammation with organ damage, it appears effective to initiate HP to reduce the inflammation [59]. Also, in the presence of AKI, a sequential HP treatment followed by CRRT (sequential approach) or a combined HP treatment integrated into the CRRT circuit (combined approach) could be considered (Figure 3) to treat AKI and CRS. According to our experience, the possibility to use a sequential approach is safer in the setting with lower experience and few personnel, as the caregivers can focus on the single treatment providing the best effectiveness and safety in terms of purification and circuit patency. The combined approach for its complexity requires personnel with experience and advanced technical skills for the management of the integrated devices, the circuits patency, and the technical problems inherent in the treatment combinations (Figure 3b). We propose a COVID-19 patient algorithm to guide the treating physician in the choice of the treatment(s) in the most common COVID-19 clinical settings of AKI and/or CRS (Figure 6). COVID-19 patients are complex and subject to rapid clinical evolution due to the organs injury; in fluid overload, CRRT can be initiated to support the lungs, to improve the pulmonary gas exchanges and heart function, achieving the best hydration status [60]. In this case, the choice of performing CRRT with absorptive filters for cytokines and endotoxin can be useful for the CRS control. Also, in this case, CRRT can be combined with or follow an HP treatment (Figure 3). As previously reported, different devices for HP require different treatment durations: to remove cytokines, CytoSorb/HA330-380 provide a continuous treatment of 24 h a day for at least 2 or 3 days, exposing the patient to a system bleeding risk due to the anticoagulation; while, to remove endotoxins, Toraymyxin for 2 h a day per 2 days exposes the patient to a reduced bleeding risk for a time window of 2 h only. As regards the clotting risk of these devices, Cytosorb/HA330-380 are more affected by an improper state of patient’s anticoagulation, as a treatment time of at least 12–24 h is required, while Toraymyxin requires a narrow anticoagulation window of 2 h that allows the patient to be anticoagulated, reducing the risk of bleeding over time. If ECMO is indicated, it is possible to further customize EBP by integrating the HP device in the ECMO circuit (CytoSorb) (combined treatment) [61,62]; while in the case of endotoxin removal it is possible to perform HP with Toraymyxin, in parallel exploiting the systemic anticoagulation required by the ECMO. If AKI is also present in a patient under ECMO, the EBP treatment can include CRRT combined to HP to remove cytokines (CRRT + HP with CytoSorb), or HP followed by CRRT (HP with CytoSorb followed by CRRT). In case of needing to remove an endotoxin, Toraymyxin can be provided before CRRT. The CRRT application with a device with absorptive capacity for cytokines, inflammatory factors, and endotoxins should also be considered, but given the rapid device saturation, it has to be replaced at least every 24 h. This treatment is simple and requires limited personnel experience, allowing to be safely provided even in ICU settings in which EBPs are rarely performed. In our experience, in the event of high bleeding risk with contraindicated systemic anticoagulation, the use of devices such as Cytosorb and Toraymyxin is challenging due to the early device clotting and the loss of treatment effectiveness. In these clinical situations, a filter with absorptive capacity, such as oXiris, in CRRT, can be used in RCA.

ECCO_2_-R [63] is another EBP available for the COVID-19 management that allows to treat hypercapnic patients with a less complex hardware than ECMO and, therefore, is usable in settings with less experience or a limited multidisciplinary approach.

In the case of malfunctioning CVCs or the need for regular patient mobilization, the need for reduced blood flows (100–120 mL/min) requires to re-evaluate the applicable EBPs to maximize their effectiveness and reduce the risk of coagulation of the devices. According to our experience, it is necessary to evaluate the maximum blood flow obtainable from the vascular access before deciding which EBP to use, as low blood flows are exposed to a higher risk of coagulation and reduced effectiveness of EBP.

In order to conciliate the reduction of blood flow and the achievement of treatment efficacy with a minimum risk of circuit clotting, the HCO-CVVHD-RCA can be used as the high cut-off filter is used in diffusive modality with low FF, allowing the use of relatively low blood flows, ensuring an appropriate depurative dose in the case of CVC dysfunctions and during patient’s mobilization. The HCO-CVVHD-RCA determines a loco-regional anticoagulation that does not alter the patient’s systemic circulation, exposing him to the bleeding risk as in the technique that requires the use of heparin or low molecular weight heparin. Finally, the low FF, the low blood flow, and the sodium citrate anticoagulation result in the lowest clotting risk. All these elements determine a longer circuit patency, a minor number of circuit alarms requiring maintenance, lower volume of hemodiafiltration fluids for the blood purification, fewer staff interventions for changing fluid bags, and finally reduced workload that is very useful in the case of lack of human resources [44].

The EBP technology can easily provide sequential or combined treatments of HP and CRRT if needed to remove endotoxin, cytokines, and inflammatory mediators, and simultaneously to treat AKI. A valid alternative to the use of specific CRRT adsorptive device could also be the use of standard filters for CRRT with less adsorbing capacity but always in regional citrate anticoagulation to maintain their patency over time. In this case, given the lower adsorptive capacity of the membranes used (AN69, Polysolfone, PMMA), high-volume CVVH allows for an increased removal of cytokines and inflammatory factors [64]. However, this convective treatment requires high volumes [65] of hemodiafiltration solutions and certainly higher blood flows to keep the FF below 20–25% [66]. Moreover, despite the RCA, the high FF required by some CVVH treatments (overall in patients with body weight higher than 80–90 kg) could lead to device clotting, in particular in the case of reduced blood flows due to CVC dysfunctions or patient mobilization. Therefore, this approach could result in continuous alarms and early circuits and filter clotting with increased workload and reduced depurative doses provided to patients.

Devices for ECCO_2_-R, according to their easy use in a similar HP modality, can be easily used by personnel involved in the management of CRRT in ICU using the nephrologist’s experience for the extracorporeal circulation management. Interestingly, a new device (Seraph 100, Fresenius Medical Care Deutschland GmbH) is promising for the reduction of pathogen SARS-CoV-2 by filtering it from the bloodstream in adjunction to the medical therapy; this device can be used in the early phase of SARS-CoV-2 damage to reduce the CRS and organs damage [67]; thanks to its characteristics, it can be used alone in HP or combined in the CRRT to support AKI patients. The possibility of having such EBP and organ support can be useful in supporting and treating COVID-19 patients [59].

Given the management complexity of EBPs and the need for specific knowledge and experience in the field of extracorporeal circulation, a multidisciplinary approach is required and the need to codify and regulate treatments by introducing protocols becomes fundamental to minimize clinical risk, especially in the lack of human recourses requiring untrained personnel use. The need to better understand the purifying characteristics of such devices in terms of molecules removal requires standardization and future studies to provide users with specific information and to confirm the device’s intended use and the real effectiveness in these patients’ population. In critical clinical settings, such as the COVID-19 ICUs, the continuous request for extra-corporeal treatments supervision should not be underestimated in term of work overload, complexity of treatments, and safety [68]. In fact, parameters, such as the weight loss per hour, have to be entered continuously into the device user interface by caregivers according to the dynamic clinical condition, leading to increased risk of cross-contamination, use of personal protective equipment, and the need for stringent and demanding protocols. In the future, the possibility of the remote control of these devices may decrease the frequency of unnecessary interventions and reduce the risk of exposure for both patients and healthcare personnel [68].

## 4. Conclusions

The variability of SARS-CoV-2 infections depends on the patient’s inflammatory state that determines a broad spectrum of diseases ranging from asymptomatic form to severe MOFS requiring the use of artificial organs and EBP. It is clear that the ICU staff and treating physicians should be familiar with the concept that EBP represents today an important lifesaving strategy in the COVID-19 management. This is important not only in the single patient treatment, but also to rationalize resources in the context of a global emergency. To prevent patients from worsening allows to save resources for treating a greater number of patients and reserving ICUs for the most critical patients. As a limitation, the techniques described in this paper were tested for sepsis, but few results achieved seem encouraging. Even if there are no standardized protocols, the familiarity and ability to perform EBP can play an essential role in tailoring the appropriate EBP for COVID-19 patients in order to design future randomized studies to analyze their efficacy and impact on patients’ outcome and mortality. For now, these techniques can be configured as a valid extracorporeal support in the critically ill patient affected by COVID-19, providing valid help in the patient’s ICU treatment according to the local skills and experience.

## Figures and Tables

**Figure 1 biomedicines-10-02017-f001:**
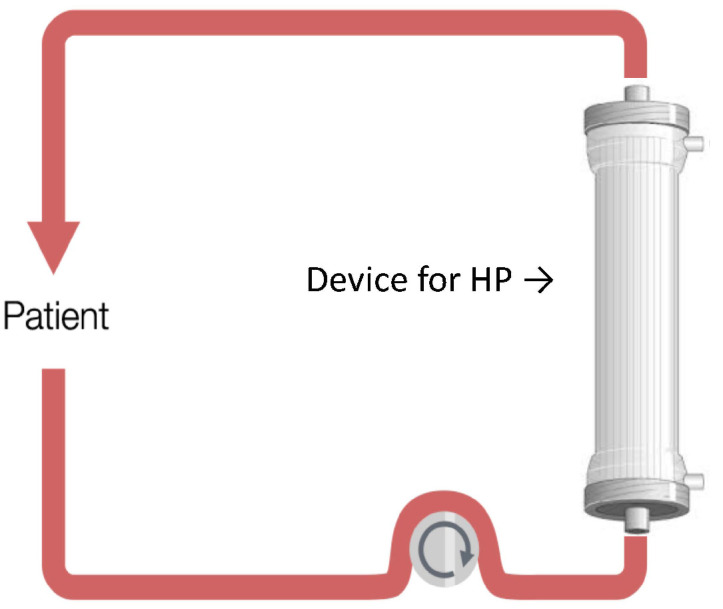
Hemoperfusion (HP).

**Figure 2 biomedicines-10-02017-f002:**
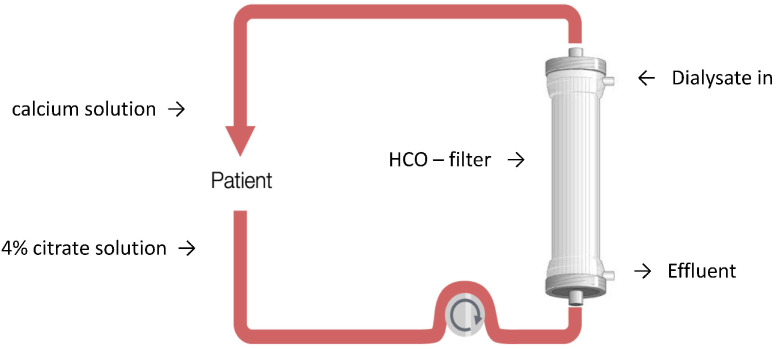
High cut-off-continuous veno-venous hemodialysis in regional citrate anticoagulation (HCO-CVVHD-RCA).

**Figure 3 biomedicines-10-02017-f003:**
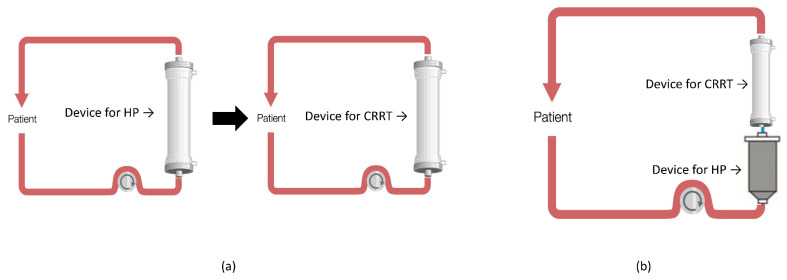
Sequential (**a**) and combined (**b**) treatments.

**Figure 4 biomedicines-10-02017-f004:**
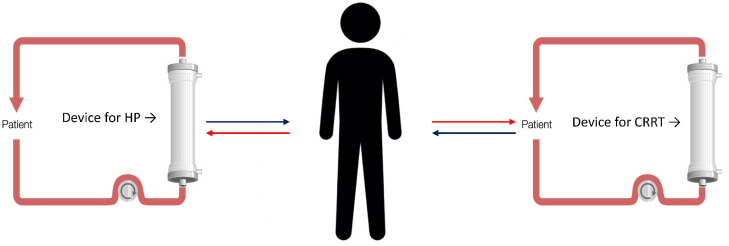
Hemoperfusion and continuous kidney replacement therapy (CRRT) with two different vascular accesses.

**Figure 5 biomedicines-10-02017-f005:**
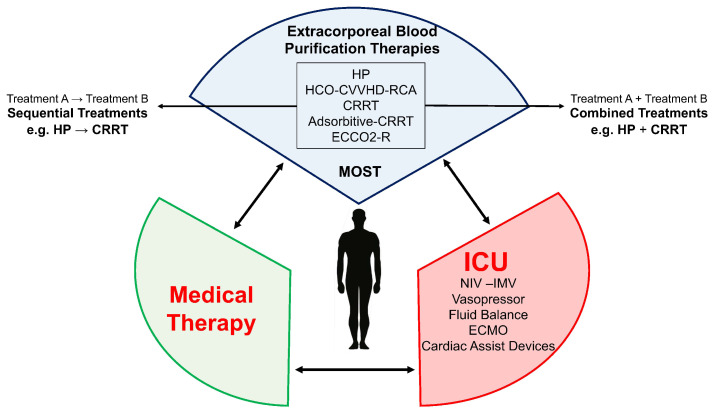
Integration of EBP in the medical therapy.

**Figure 6 biomedicines-10-02017-f006:**
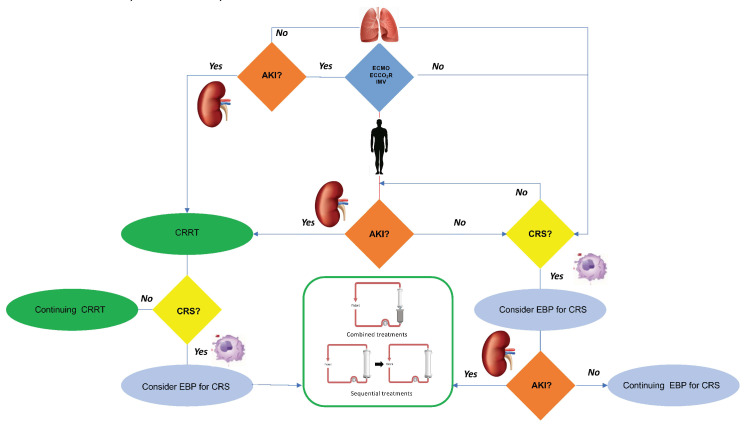
Algorithm to guide in the extracorporeal blood purification treatment in COVID-19 patients.

**Table 1 biomedicines-10-02017-t001:** EBPs comparison.

Device	Manufacturer	Composition	Device Type	Specificity If Removal	Target in COVID-19	Treatment Type	Blood Flow (mL/min)	Anticoagulation	Duration of Single Device	Use with Other Treatment
**CytoSorb**	CytoSorbents Corporation, Princenton, NJ, USA	beads in polystyrene divinylbenzene copolymer with a biocompatible polyvinylpyrrolidone coating	hemoadsorber	non-selective capacity	cytokines and inflammatory mediators	HP	150–500 mL/min (maximum flow 700 mL/min) with a minimum of 100 mL/min	Heparin; aPTT between 60 and 80 s (or ACT of 160–210 s)	24 h	CRRT/ECMO
**HA-330**	Jafron Biomedical Company, Zhuhai, China	neutro-macroporous resin adsorbing beads in non-ionic styrene divinylbenzene copolymers	hemoadsorber	non-selective capacity	hydrophobic or protein-bound exogenous substances, cytokines, protein-bound uremic toxins, middle uremic toxins, free hemoglobin, and myoglobin	HP	100–250 mL/min	Heparin; desired aPTT between 60 and 80 s (or ACT of 160–210 s)	24 h	CRRT/ECMO
**Toraymyxin**	Toray Industries Ltd., Tokyo, Japan	polymyxin B-immobilized on polystyrene derivative fibers	hemoadsorber	selective capacity	endotoxin (direct adsorption of inflammatory mediators, cytokines, and the activated monocytes and neutrophils apheresis)	HP	100–120 mL/min	Heparin; desired aPTT between 60 and 80 s	2 h	-
**Septex**	Baxter, Round Lake, IL, USA	polyarylethersulfone membrane of 1.1 m^2^	High Cut-Off filter for CVVHD	non-selective capacity	cytokines and inflammatory mediators	CVVHD in RCA or with Heparin	80–200 mL/min	Trisodium citrate or heparin	72 h	-
**Emic-2**	Fresenius Medical Care, Bad Homburg, Germany	polysulfone membrane of 1.8 m^2^	High Cut-Off filter for CVVHD	non-selective capacity	cytokines and inflammatory mediators	CVVHD in RCA or with Heparin	100–200 mL/min	Trisodium citrate or heparin	72 h	-
**oXiris**	Baxter, Round Lake, IL, USA	acrylonitrile and sodium methallyl-sulfonate-copolymer and as surface treatment agent polyethyleneImine (PEI) and heparin	Filter for all CRRT	non-selective capacity	endotoxins, cytokines, and inflammatory mediators	CRRT in RCA or with Heparin	80–200 mL/min in RCA	Trisodium citrate or heparin	72 h	-
							120–200 mL/min with Heparin			

## Data Availability

Not applicable.

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
