# Peer review of "The Supporting Role of Combined and Sequential Extracorporeal Blood Purification Therapies in COVID-19 Patients in Intensive Care Unit"

_biomedicines, 2022, doi:10.3390/biomedicines10082017_

Round 1
Reviewer 1 Report
Dear Editor
Thank you for having the opportunity to review this interesting manuscript.
On the other hand, being a review, I would have expected to find some indications in the manuscript that I believe are missing.
Various methods of extracorporeal blood purification are described in great detail. In the literature, what are the case studies and what are the evidence of effectiveness in patients treated for COVID-19? As part of a hemodialysis unit, do the authors have any experience with this? In fact, there is no link between what is announced in the title and what appears in the manuscript. Referring to COVID-19, I would have expected to find comparative tables in the effectiveness of the various treatments on severe syndromes caused by advanced disease. For example, what is the effectiveness in Cytokines Release Syndrome? Which is preferable? And so on.
Are the authors able to implement the manuscript by providing evidence of effectiveness?
Minor
In my opinion, the references in the text and in the reference list do not comply with the editorial rules.
Author Response
Reviewer 1
Thank you for having the opportunity to review this interesting manuscript.
On the other hand, being a review, I would have expected to find some indications in the manuscript that I believe are missing.
We included indications and suggestions in the discussion according to our experience and the litterarure.
Various methods of extracorporeal blood purification are described in great detail.
In the literature, what are the case studies and what are the evidence of effectiveness in patients treated for COVID-19?
Unfortunately, multicentric or randomized studies have not yet been published in the literature giving indications on the efficacy of such treatments. There are only pilot studies or single-center studies that analyze a small number of patients with non-definitive and sometimes conflicting results. These results derive from the difficulty in recruiting patients with the same characteristics (severity of COVID-19, comorbidities, etc.) that can make the studies homogeneous and comparable for patients’ population. Local policies for the indications and timing of initiation of these treatments are affected by factors such as clinical experience, the presence of a multidisciplinary team, the ability to provide treatments and local economic resources; this is can be a bias for the studies. All these elements make the available efficacy data not comparable at present. Based on case reports and pilot studies, the use of these treatments could be useful in patients affected by COVID-19, but certainly a standardization of the indications, timing and methods of these treatments will be necessary in the future to perform further studies on homogeneous patient populations and statistically significant numbers. The purpose of this paper is to illustrate how to perform EBPs, providing a practical support to extracorporeal therapies in COVID-19 patients with AKI and/or CRS. We think that this is the first step to perform these treatments in COVID-19 patients and to apply treatments protocols to assess their efficacy by multicentric and randomized studies in a statistically correct population.
As part of a hemodialysis unit, do the authors have any experience with this?
We have included indications and suggestions in the discussion according to our experience.
In fact, there is no link between what is announced in the title and what appears in the manuscript. Referring to COVID-19, I would have expected to find comparative tables in the effectiveness of the various treatments on severe syndromes caused by advanced disease. For example, what is the effectiveness in Cytokines Release Syndrome? Which is preferable? And so on.
We rewrote the title in order to make the content of the review adhere to the title. The treatment table has been improved and an algorithm has been included for deciding which EBP to start in the individual patient.
Are the authors able to implement the manuscript by providing evidence of effectiveness?
We have reworked the text to improve the paper according to the reviewer's instructions.
Minor
In my opinion, the references in the text and in the reference list do not comply with the editorial rules.
We re-edited the references according to the instructions of the journal (we used endnote with the correct MDPI template).
Reviewer 2 Report
The German and English nomenclature is mixed up e.g. CKRT in the body text vs CRRT in the figure legends.
Abbreviation CRS is misleading as it may be cardiorenal syndrome as well in this context.
Page 2 Lines 66-67: "prevent further AKI and control the volume overload that may deteriorate lung exchanges"... Please indicate what this means properly. Do Authors mean lung transplant? For sure no.
Page 2 Lines 90-92, sentence is confusing: " Much of the current experience on MOST has been gained upon KRT connected to ECOS in order to perform CKRT in combined or sequential therapy with special devices for ECMO, extracorporeal carbon dioxide removal or Hemoperfusion (HP)." Please refrase.
Page 3: Line 129: "Cytosorb is an option to treat COVID-19 [28] [23] in ICU with the objective of reducing the “cytokine storm” [29]". This is only a possible adjunct, but not an universal method for COVID.
Page 4: Line 145 „heparin of 50-70 IU/Kg and a maintenance dose of 15-20 IU/Kg should be evaluated” The maintenance dose should be corrected.
Page 4 Line 148: „central venous catheter (CVC) performance and the device use in HP alone”. Please define properly what performance needs.
Please give the effective minimal diameter and flow rate to achieve an effective purification with each modality.
Page 5 Lines 243-245. „Blood flow may be low enough (100- 200 mL/min) to prevent extracorporeal circulation malfunction in case of CVC dysfunction or inadequate blood flow for patient mobilization”.
Please refrase, the sentence has no clear meaning.
Page 6 Line 247-248” excluded from the use in Continuous Veno-Venous Hemofiltration (CVVH) due to possible albumin losses”. Please indicate the mechanism of albumin loss, and specify the same for all the modalities and membrane types.
Page 6 line 276 conccerning oXiris: „the immobilized heparin on membrane surface gives an anti-thrombogenic property”. Please indicate the duration while heparin is really „anchored „ on the surface of the membrane , in the other way: how long one does not have to use iv UFH?.
Line 293: please specify the mechanism of HIT when the membrane is covered by „immobilized heparin”.
The discussion should be reworked since it overhelms the reader wwith a lots of configurations, modalities. A decision algorythm should be provided to guide the reader wich technology would be the best for specific deiseases as blood purification and additional disease-specific adjuncts. Should the modalities didactically hierarchized cocerning the risk of potential thrombogenic and bleeding adverse events, and efficacy and minimal time of treatment modalities.
Lines 400-410 contain absolutely general but not specific conclusive remarks. Should be rewrited.
Figures 5a,b are very general. Please indicate more thoroughly the differences.
Author Response
The German and English nomenclature is mixed up e.g. CKRT in the body text vs CRRT in the figure legends.
The CKRT is reported for all paper.
Abbreviation CRS is misleading as it may be cardiorenal syndrome as well in this context.
Cytokine release syndrome (CRS) is also correctly reported in PubMed, we think that it can be used in paper. Readers can find the abbreviation in the text after the Cytokine Release Syndrome was introduced with the abbreviation CRS and therefore have no problems.
Page 2 Lines 66-67: "prevent further AKI and control the volume overload that may deteriorate lung exchanges"... Please indicate what this means properly. Do Authors mean lung transplant? For sure no.
We better explained the sentence in the text. With “lung exchanges” we mean gas lung exchange.
Page 2 Lines 90-92, sentence is confusing: " Much of the current experience on MOST has been gained upon KRT connected to ECOS in order to perform CKRT in combined or sequential therapy with special devices for ECMO, extracorporeal carbon dioxide removal or Hemoperfusion (HP)." Please refrase.
Much of the current experience on MOST has been gained upon the KRT connected to ECOS. The evolution of KRT in CKRT combined or in sequential modality with special devices can better support patient under ECMO, extracorporeal carbon dioxide removal or Hemoperfusion (HP).
Page 3: Line 129: "Cytosorb is an option to treat COVID-19 [28] [23] in ICU with the objective of reducing the “cytokine storm” [29]". This is only a possible adjunct, but not an universal method for COVID.
We agree with the review: the Cytosorb hemoperfusion is not an universal method to treat COVID, it can support and treat patient with very important inflammation. In the text we used this sentence as we are presenting this device according to the available literature and the FDA Authorization to use the device to treat COVID-19 patient. To better explain the text we insert a rewritten sentence in the paper.
Cytosorb is an option to support and treat COVID-19 [28] [23] in ICU with the objective of reducing the “cytokine storm” [29], receiving the FDA Emergency Use Authorization on April 10, 2020.
Page 4: Line 145 „heparin of 50-70 IU/Kg and a maintenance dose of 15-20 IU/Kg should be evaluated” The maintenance dose should be corrected.
The heparin bolus and maintenance doses are reported from the technical data sheet of the device from de manufacturer (see the original part of this document below)
Page 4 Line 148: „central venous catheter (CVC) performance and the device use in HP alone”. Please define properly what performance needs.
Please give the effective minimal diameter and flow rate to achieve an effective purification with each modality.
The Cytosorb 300 blood flows for technical data sheet can range from 100 to 700 mL/min according to the vascular access that can be used. With standard CVC for hemodialysis the range is around 200-300 mL/min, if the Cytosorb is placed in the ECMO circuit, it can reach the blood flow of 500-600 mL/min. In standard condition in ICU the standard CVC for hemodialysis is used to obtain the extracorporeal circulation. We insert in the text a new sentence to better explain this point.
In order to achieve an effective purification with all these modalities of blood purification is required at least a standard temporary 12 F double lumens CVC catheter for hemodialysis that can guarantee a blood flow up to 300 ml / min.
Page 5 Lines 243-245. „Blood flow may be low enough (100- 200 mL/min) to prevent extracorporeal circulation malfunction in case of CVC dysfunction or inadequate blood flow for patient mobilization”.
Please refrase, the sentence has no clear meaning.
We rewrote the sentence according the suggestions.
In this technique the required blood flow for purification is lower thus it may be set low enough (100- 200 mL/min) to prevent extracorporeal circulation malfunction in case of CVC dysfunction that not allow higher blood or in case of inadequate blood flow due to patient mobilization.
Page 6 Line 247-248” excluded from the use in Continuous Veno-Venous Hemofiltration (CVVH) due to possible albumin losses”. Please indicate the mechanism of albumin loss, and specify the same for all the modalities and membrane types.
We insert a better explanation of the albumin loss in the text.
With HCO-CVVHD-RCA both Septex and EMIC-2 are used in CVVHD, while they are excluded from the use in Continuous Veno-Venous Hemofiltration (CVVH) due to possible albumin losses. In fact, with filter with MCO or HCO, in the presence of convective therapies, such as CVVH and CVVHDF, the convective flow can cause a significant loss of albumin for its entrainment through the pores of these membranes while this fact is not present in diffusive treatments such as CVVHD also using MCO or HCO. Furthermore, the CVVHD treatments has low filtration fractions (FF), as it is based on diffusion only allowing the use of relatively low blood flows while ensuring the septic depurative dose [46].
Page 6 line 276 conccerning oXiris: „the immobilized heparin on membrane surface gives an anti-thrombogenic property”. Please indicate the duration while heparin is really „anchored „ on the surface of the membrane , in the other way: how long one does not have to use iv UFH?.
In the oXiris filter, heparin is covalently bound to the filter surface and not adsorbed. There is no release of heparin over time (theoretically). The heparin on the surface of the filter acts by binding the various factors that can interact.
This is also reported in the technical data sheet of the product.
Line 293: please specify the mechanism of HIT when the membrane is covered by „immobilized heparin”.
We believe that the description of HIT type 2 is not the subject of this paper and that it should not be included in the text. In any case, the use of Oxiris is contraindicated by the technical data sheet in case of known HIT type 2.
In the event of the presence of anti PF4 antibodies there will be a systemic reaction in relation to the possible loss of heparin from the filter.
The discussion should be reworked since it overhelms the reader wwith a lots of configurations, modalities. A decision algorythm should be provided to guide the reader wich technology would be the best for specific deiseases as blood purification and additional disease-specific adjuncts. Should the modalities didactically hierarchized cocerning the risk of potential thrombogenic and bleeding adverse events, and efficacy and minimal time of treatment modalities.
The discussion was reworked to simplify the concepts inherent in the treatment modalities of extracorporeal blood purification. A decision algorithm has been developed to govern the reader on which technology would be the best in the treatment in relation to the presence of AKI / CRS and to local facilities. A comment was presented regarding the risk of circuit coagulation, the bleeding risk in patients, and the minimum time of application of the specific thecniques (see also table 1).
Lines 400-410 contain absolutely general but not specific conclusive remarks. Should be rewrited.
The text has been written in relation to what was indicated by the auditor.
Figures 5a,b are very general. Please indicate more thoroughly the differences.
Since the techniques CVVH, CVVHD, CVVHDF with and without anticoagulation or in regional anticoagulant citrate are well known to all physicians working in ICU, we believe that the figure can be completely removed from the text.
Round 2
Reviewer 1 Report
The authors modified the manuscript taking into account my suggestions. I have nothing else to add.
Author Response
Since no further requests for revision have been made, no further clarifications need to be made.

Reviewer 2 Report
CRRT is adviced to use instead of CKRT. It is the official abbreviaton.
"gas lung exchange" should be replaced with "pulmonary gas exchange.
"...very important inflammation..." should be corrected, this has no exact meaning.
Page 4. concerning UFH dasage for maintenance: still, if a long term CRRT modality is used, a single bolus for maintenance is not enough, dose over time period should be defined. efg. IU/Kg/h.
Page 5 , lines 243-245: still should be clarified and English sentence should be checked. "... in case of CVC dysfunction that not allow higher blood or in case of... "
Line 293 (cpncerning HIT-2 and PF 4) : please state whether it is or is not to be deleted.
Fig 5a, b is considered by the authors that they "can be completely removed from the text. " Please state whether to be deleted or not.
Author Response
CRRT is adviced to use instead of CKRT. It is the official abbreviaton.
We used the CRRT word in the paper.
"gas lung exchange" should be replaced with "pulmonary gas exchange”.
We replaced "gas lung exchange" with "pulmonary gas exchange”.
"...very important inflammation..." should be corrected, this has no exact meaning.
We used only the word “inflammation”.
Page 4. concerning UFH dosage for maintenance: still, if a long term CRRT modality is used, a single bolus for maintenance is not enough, dose over time period should be defined. efg. IU/Kg/h.
we have included the maintenance heparin dose in the text.
Page 5 , lines 243-245: still should be clarified and English sentence should be checked. "... in case of CVC dysfunction that not allow higher blood or in case of... "
In this technique the required blood flow for purification is lower thus it may be set low enough (100- 200 mL/min) to prevent extracorporeal circulation malfunction in case of CVC dysfunction not allowing higher blood flow or in case of inadequate blood flow due to patient mobilization.
Line 293 (concerning HIT-2 and PF 4) : please state whether it is or is not to be deleted.
We deleted it.
Fig 5a, b is considered by the authors that they "can be completely removed from the text. " Please state whether to be deleted or not.
The figure has been deleted as it did not provide useful information to the reader.
Round 3
Reviewer 2 Report
EVerything is coerrected as per review. Thank you